# COVID-19 Vaccination and Medical Liability: An International Perspective in 18 Countries

**DOI:** 10.3390/vaccines10081275

**Published:** 2022-08-07

**Authors:** Flavia Beccia, Maria Francesca Rossi, Carlotta Amantea, Leonardo Villani, Alessandra Daniele, Antonio Tumminello, Luna Aristei, Paolo Emilio Santoro, Ivan Borrelli, Walter Ricciardi, Maria Rosaria Gualano, Umberto Moscato

**Affiliations:** 1Section of Hygiene, University Department of Life Sciences and Public Health, Università Cattolica del Sacro Cuore, 00168 Rome, Italy; 2Section of Occupational Health, Department of Woman and Child Health and Public Health, Università Cattolica del Sacro Cuore, 00168 Rome, Italy; 3Department of Law, LUISS Guido Carli University, 00198 Rome, Italy; 4Department of Woman and Child Health and Public Health, Fondazione Policlinico Universitario A. Gemelli IRCCS, Largo Francesco Vito 1, 00168 Rome, Italy; 5Department of Health Science and Public Health, Università Cattolica del Sacro Cuore, Largo Francesco Vito 1, 00168 Rome, Italy; 6Department of Public Health Sciences, University of Turin, 10124 Turin, Italy

**Keywords:** compensation programs, COVID-19 vaccination, legal system, legislation, medical liability

## Abstract

The COVID-19 vaccination has proven to be the most effective prevention measure, reducing deaths and hospitalizations and allowing, in combination with non-pharmacological interventions, the pandemic to be tackled. Although most of the adverse reactions to vaccination present mild symptoms and serious effects are very rare, they can be the cause of legal action against the healthcare workers (HCWs) who administered it. To highlight differences in the medical liability systems, we performed a search for the three most populous countries in each continent on vaccine injury compensation programs, new laws or policies to protect HCWs administering vaccinations introduced during the COVID-19 pandemic, and policies on mandatory vaccinations, on literature databases and institutional sites. We found that in seven countries the medical liability system is based on Common Law, while in eleven it is mainly based on Civil Law. Considering the application of specific laws to protect HCWs who vaccinate during the pandemic, only the USA and Canada provided immunity from liability. Among the countries we analyzed, fourteen have adopted compensation funds. From an international perspective, our results highlight that in eleven (61.1%) countries medical liability is mainly based on Civil Law, whilst in seven (38.9%) it is based on Common Law.

## 1. Introduction

Since the first half of 2020, the health emergency, which has seen the entire planet involved in the management of COVID-19 patients, has led to great stress for healthcare workers (HCWs) due to increased workload [1,2]. The HCWs involved in the emergency management network were the pillars on which the response to the COVID-19 outbreak were based. Therefore, it is crucial to preserve as much as possible their physical and mental health as well as to guarantee them legal and regulatory protection [3]. Indeed, especially during the first period of the pandemic and in the field of COVID-19 vaccinations, HCWs had to work under conditions of extreme scientific uncertainty, having to decide in some cases between their own legal safety and the protection “at their own risk” of the health of patients. Although the COVID-19 vaccination being one of the most effective and safest measures to prevent infection, hospitalization, and death, the vaccination coverage remains low in most of the countries [4,5,6]. Indeed, fake news and miscommunication fueled vaccine hesitancy and distrust of vaccination and vaccination physicians [7]. Moreover, as with any other medical treatment and vaccine, COVID-19 vaccination is not free from adverse reactions. Although most of these reactions present a mild symptomatology and severe effects are very rare, these may be the cause of legal actions against the healthcare operator who administered them [8]. In order to increase the vaccination coverage and the trust in COVID-19 vaccination, both national and international institutions promoted vaccine injury compensation programs (VICPs), while specific laws provide physician immunity from injury’s liability caused by vaccine administration. In Europe, for example, France and Italy have resorted to these instruments, establishing a specific compensation system for COVID-19 vaccine-related injuries for the former and a criminal shield for the latter [9,10]. At international level, the World Health Organization (WHO) and the United Nations promoted the COVAX No-Fault Compensation Program, an example of VICPs to deliver safe and effective COVID-19 vaccines to high-risk and vulnerable populations [11].

In this context, the purpose of our research is to identify, among the three most populous countries on each continent (Africa, Asia, Europe, North America, Oceania, and South America), which medical liability system is in place, and if during the COVID-19 pandemic there have been introduced VICPs or laws or new policies to protect vaccinating HCWs, providing also information whether, in the selected countries, vaccination is mandatory for specific occupational groups or the general population.

## 2. Materials and Methods

### 2.1. Countries Selection

In order to perform the research on a global scale, for reasons of feasibility, we decided to include the three countries with the highest population for each continent, considering North and South America separately. Therefore, we searched the United Nations database [12], and extracted the countries to include. Data were updated as of March 2022 for all the 18 selected countries.

### 2.2. Medical Liability Research

We searched on PubMed for articles in English published up to May 2022 on medical liability, with a focus on laws or policies implemented during the COVID-19 pandemic for vaccinating HCWs. The following query was used, tailoring it for each country adding it as follows on an ad hoc basis: ((“medical liability” OR “medical negligence” OR “medical malpractice”) AND (law* OR legislation OR policy OR policies) AND (“Country name”)). The articles were first screened according to title and abstract and then the full texts of eligible articles were evaluated. Moreover, using the same search query, a grey literature research was performed in English on the Google search engine, retrieving articles focusing on medical liability with particular attention to policies and laws implemented during the COVID-19 pandemic. Finally, we searched each country’s institutional repositories for additional information. We combined the results from the three different sources to outline the framework of medical liability in each country, emphasizing, where appropriate, the peculiar changes that occurred during the pandemic and related to vaccinating doctors.

### 2.3. Vaccination Coverage, Mandatory Vaccination and Compensation Programs

Vaccination doses and coverage were extracted from the World Health Organization database [13], on 11 March 2022, two years from the beginning of the pandemic [14,15], and reported as people fully vaccinated per 100 population, as a raw frequency and percentage. Moreover, we searched each country’s institutional repositories and websites to find information about the mandatory vaccination policies, for the public or in specific categories, and compensation programs adopted or specifically developed by each state. We reported the key findings through a set of country-specific profiles that outline any presence of legislative frameworks towards medical liability and policies for vaccinating HCWs or VICP.

## 3. Results

We analyzed the medical liability system in 18 countries worldwide and the introduction of laws or new policies to protect vaccinating HCWs or VICP.

The selected countries for each continent are:Africa: Nigeria, Ethiopia, and Egypt;Asia: China, India, and Pakistan;Europe: Russian Federation, Turkey, and Germany;North America: United States of America, Canada, and Mexico;Oceania: Australia, Indonesia, and Papua New Guinea;South America: Brazil, Colombia, and Argentina.

Of these countries, only two introduced specific policies on COVID-19 and medical liability (Canada and USA). Eight countries declared COVID-19 vaccination mandatory for specific sub-populations (i.e., HCWs, employees, and teachers) (Nigeria, Egypt, Russian Federation, Turkey, USA, Canada, Australia, and Indonesia). The vaccination coverage was highly variable between the countries, from 4.1% vaccinated in Nigeria to 84% in China. Ten of the studied countries were included in the COVAX No-Fault Compensation Program, sponsored by the World Health Organization (WHO) and the United Nations (Argentina, Brazil, Colombia, Egypt, Ethiopia, India, Indonesia, Nigeria, Pakistan, and Papua New Guinea), while four countries adapted the existing compensation programs to COVID-19 (Germany, USA, Canada, and Australia). Four countries do not have any program in place (China, Russian Federation, Turkey, and Mexico). All our findings regarding medical liability, vaccination coverage, mandatory vaccination, and compensation programs are reported in detail in Table 1.

### 3.1. Africa

#### 3.1.1. Nigeria

There are four distinct legal systems in Nigeria, which include English Law, Common Law, Customary Law, and Sharia Law [16]. The duties and obligations of medical practitioners can be founded on Common Law, Statutory Law, and ethical obligations [17]. The principal law regulating the medical profession in Nigeria is the Medical and Dental Practitioners Act (MDCN) 2004 [18]. In 2008, this MDCN codified the rules of professional conduct for medical and dental practitioners in the Nigerian Code of Medical Ethics. Under the Criminal Code, the criminal liability of a medical practitioner for the negligent treatment of a patient is based on gross breach of duty which the medical practitioner owes the patient. According to the Code, there is negligence in cases of omission to perform a duty or to carry out an act. The Code also provides for liability for malicious service contract breaking, if such will endanger human life or cause serious bodily harm [19].

#### 3.1.2. Ethiopia

In Ethiopia, the medical liability system is mainly based on Civil Law. The 1960 Civil Code recognized both contractual and non-contractual liabilities of medical institutions. In cases of extra-contractual liabilities, the Civil Code provides no separate rules that specifically deal with medical malpractices or negligence. Instead, the general tort provisions become applicable for medical malpractice actions. Conversely, the Civil Code enumerates some provisions regarding the contractual dimension of medical institutions’ liability. Specifically, article 2651 of the Civil Code provides the medical institutions that shall be liable for the damage caused to a sick person if the fault is committed by physicians or auxiliary staff that the institution employs. As such, the article excludes the liability of medical institutions for the damages caused by an independent contractor or non-employee physicians. Indeed, the liability of medical institutions in Ethiopia depends upon the existence of an employment relationship between the physician and the institutions [20].

#### 3.1.3. Egypt

In Egypt, medical professionals who commit errors that cause injury or death in patients are civilly liable according to Civil Code No. 131/1948 [21,22]. They are also subject to criminal investigation and penalties under Penal Code No. 58/1937. In Egypt, the cornerstone of tortious liability is Article 163 of the Civil Code, which states that “every error resulting in damages to a third party requires compensation from the liable party”. Tortious responsibility is hence composed of three elements: the error, the damage, and the causal relationship between these factors. In medical malpractice, the simplest element to ascertain is the damage [23]. However, given the unpredictability and variances in acceptable treatment methods, it is difficult to differentiate between medical error and prognostic uncertainty or inevitability. Individuals are not responsible under Egyptian law if the damage resulted from a negative externality beyond their control [24]. As of now, medical torts are a sub-category of civil torts. However, a draft Medical Liability Law by the Egyptian Medical Syndicate has been circulated through Parliament recently. Ideally, there would be only one category of medical tortious responsibility, and the civil and criminal penalties would differ in accordance with the gravity of the error and the risk of the medical procedure [25].

### 3.2. Asia

#### 3.2.1. China

In China, the history of medical malpractice liability originates in the 1990s as a fault-based tort liability (Article 106 of the 1986 General Principles of Civil Law). In 2009, the Chinses legislator stipulated the Tort Liability Law by adopting an objective standard of fault under certain circumstances. On May 2020, the medical malpractice liability was codified with minor modifications in the Book VII of the Chinese Civil Code, which consists of 11 articles, from Article 1218 to 1228 (Chapter 6). This new law superseded the old regulation that required the defendant to prove the absence of causality. In malpractice of medical products and blood transfusion, the physician will assume non-fault liability. In addition, unlike most of the other jurisdictions, Chinese law makes the medical facility liable for the damage suffered by the patient, instead of the physician or other health practitioners [26].

#### 3.2.2. India

In India medical liability depends on Common Law. The doctors have a duty of care to patients and any failure to fulfil that duty makes the medical professional liable by law. Negligence makes the doctors liable under criminal law. There are many statutes regulating aspects of the medical profession and if the patient desires, he/she may proceed in the Criminal Court (if the seriousness of the case permits) and at the same time in the consumer commission of the Civil Court. About the civil remedy, if the intent of the aggrieved patients is to seek damages, they can choose between Jurisdictional Civil Court or Consumer Protection Tribunal. In 1996, the Supreme Court declared that the Consumer Protection Act (CPA) is also applicable to doctors, transforming de facto the patient into a “consumer”. The inclusion of medical professionals (or services) into the ambit of CPA was the first attempt, which encouraged many dissatisfied patients to use legal remedy. However, to argue under the CPA, patients must have paid for the medical service, while if a medical service is available free of cost or as charity, the only way forward is through the Civil Courts [27].

#### 3.2.3. Pakistan

The Pakistani medical liability model is based on Common Law, while negligence is usually covered up by administrators of private and public hospitals. The main responsibility to penalize doctors who have been negligent in their practice rests with the Pakistan Medical and Dental Council (PMDC), authorized by its Constitution to initiate an inquiry in front of the Medical Tribunal. The Council can punish, suspend, or even revoke the license to practice healthcare of those professionals who have been found guilty of violating the code of medical conduct. Lawsuits can be brought against medical negligence under the national criminal, tort, and consumer law. Under the 1973 Constitution of Pakistan, it is also established that no one, including doctors, enjoys complete immunity [28]. Hence, there is no system of rules that objectively regulates medical liability.

### 3.3. Europe

#### 3.3.1. Russia

Criminal liability of a medical professional who has improperly performed professional duties is regulated by several articles of the Criminal Code of the Russian Federation, depending on the type of consequences (grave injury to health—Article 118; or death of a patient—Article 109). Criminal medical liability includes the improper performance of professional duties by a medical practitioner that results in a patient’s death or grave injury to his/her health.

However, the most common type of legal liability for medical organizations and medical professionals is civil liability. The main principle of civil liability is full compensation for the damage caused, as extensively determined by the provisions of Chapter 59 of the Russian Civil Code.

Since the Civil Code establishes that the employer shall compensate for damages caused by its employee, medical malpractice suits are brought against a medical organization. However, the causal link between the conduct of the medical professional and the resulting consequences for the patient must always be proven, while the employer is civilly liable no matter if the liability of the employee who caused the damage is criminal or administrative. Regardless of the existence of a medical treatment contract, medical liability is generally subsumed under non-contractual civil responsibility. Article 54 governs the ordinary cases of medical maltreatment, while Article 55 concerns the incorrect medical disclosure [29,30].

#### 3.3.2. Germany

Medical liability in Germany is imputed on the commitment of errors in treatment and/or on the obtainment of informed consent, without a distinction between contractual and tortious liability. Indeed, both errors can involve a contractual liability based on a treatment contract pursuant to §280, paragraph 1, sentence 1, of the German Civil Code, as well as a tortious liability independent of such a contract pursuant to §823, paragraph 1, of the German Civil Code [31]. The prerequisite is that the medical error has causality for the injury to the patient’s life, body, or health. The physician or surgeon is liable for bodily injury or wrongful death if the failure to provide proper care proves detrimental or, worse, fatal to the patient’s health. Aside from this liability, a physician’s responsibility may be based on a professional service contract if the physician or surgeon in question has contractually promised medical treatment [32]. In court practice, as well as in the legal literature, it is established that a medical practitioner, even within the framework of a contract of services, is only obligated to perform standard professional treatment and not to cure the patient [33]. In 2013, the “Patient’s Rights Law” became effective in Germany. This law did not significantly change the pre-existing law, but improved enforcement of the rights of the patients, stating the requirements for informed consent and treatment [31].

#### 3.3.3. Turkey

In Turkish law, medical malpractice is regulated on the general principles of civil, penal, and in some cases administrative responsibilities. However, medical malpractice liability in Turkey rests almost entirely within the fault-based civil tort liability system and civil contract law, with very specific and discreet areas of criminal and strict liability. The latter cases of responsibility are attributed if crimes against bodily integrity occur, including voluntary or involuntary manslaughter as well as negligent and felonious bodily injury. Under civil tort liability, the injured plaintiff has the burden of persuasion and production and can only obtain compensation if the healthcare provider was negligent. The plaintiff must prove the loss, the causal link between this loss and a fault, and the fault (Turkish Obligation Code, Article 50). Torts include negligent acts, such as medical malpractice. In this regard, the Turkish Obligation Code of Article 49 encompasses the compensation of the caused loss [34].

In civil contract law, the commitment of the practitioner to give his or her patient conscientious and attentive care is stated, and it is in line with the Patient Rights Regulation for Turkey [34].

### 3.4. North America

#### 3.4.1. USA

In the USA, medical malpractice law derives from the English Common Law, and it is under the authority of the individual states and not the federal government [35]. In particular, a patient may pursue a civil claim against HCWs proving four legal requirements: the existence of a legal duty of the HCW (establishment of a relationship between the patient and the HCW); the violation of the standard of care by the HCW; a causal relationship between such a breach and the patient’s injury; and the possibility for the legal system to provide redress of the damages [36]. Thus, negligence may result in civil action by the injured party. In rare cases, whether negligence occurs because of carelessness, the HCW may be subject to a charge of criminal negligence [37].

During the COVID-19 pandemic, the approval of the Public Readiness and Emergency Preparedness (PREP) Act provides liability immunity for activities related to medical countermeasures against COVID-19. Thus, the declaration provides immunity from liability for claims of loss caused by, arising out of, relating to, or resulting from the administration or the use of medical countermeasures such as diagnostics, treatments, and vaccines [38,39,40].

#### 3.4.2. Canada

The Canadian medical malpractice law follows the same principles of the US law system. Indeed, it is based on the Common Law system, which applies to all provinces and territories in Canada except for the Québec province, characterized by its own legal principles [41]. Thus, also in this case four legal requirements are asked for any legal action based on a claim for negligence: the presence of a physician–patient relationship that imposes the duty of care; the violation of the standard of care (breach of duty); the demonstration that the breach has caused the injury; and the causal connection between the breach and the injury (relationship between the harm and actions of the physician).

Several Canadian provinces implemented the legislation to create for employers and HCWs a system of liability protection against COVID-19. These legislations, known as “Bills”, provided HCWs a liability shield for damages caused by COVID-19 (spread, exposure, or otherwise), assuming they followed public health orders, guidance, and preexisting legislation [42,43,44].

During the COVID-19 pandemic, the Canadian government approved the Vaccine Injury Support Program (VISP). VISP is a no-fault program that ensures compensation to people who have experienced a permanent and serious injury because of receiving a COVID-19 vaccination.

#### 3.4.3. Mexico

Medical liability is mainly governed by Civil law. Albeit all the 32 states have their own code, Civil Law is laid down in the Federal Civil Code of Mexico. Medical liability covers patients’ damages caused by negligence, inexperience, or deceit. To resolve conflicts that arise between patients and healthcare providers, the Mexican Government established the Comisión Nacional de Arbitraje Médico. This institution tries to reconcile conflicts arising from healthcare services due to probable acts or omissions resulting from the service provisioning, probable cases of negligence (abandonment, neglect), denial of service, technical error, medical negligence, recklessness, and inexperience (lack of knowledge of technique, experience, skill), which all have consequences on the health of the patient [45,46,47,48]. Thus, under Mexican law, there is no specific procedure or system for the compensation of possible injuries or damages: patients affected or damaged by medicine or medical device may file a lawsuit (ordinary civil procedure) to request the compensation of damages.

### 3.5. Oceania

#### 3.5.1. Australia

Medical negligence in Australia follows the Common Law pattern. Monetary compensation is given to the patient if negligence is proven, meaning the plaintiff needs to prove duty, breach, and causation of damage. This must not be too remote (meaning, the damage must be under the scope of liability). Australia’s High Court defines a physician’s “duty of care” to his or her patients as diagnosis (an ongoing duty, not limited to a single case), treatment (including follow-ups if necessary), and information about material risks (the physician is only responsible if the patient did not agree to the procedure if he or she had been warned of the risk). As established by the Civil Liability Act of 2003, Section 2.1, “a doctor is not negligent if he or she acts in accordance with what was widely accepted by peer professional opinion by a significant number of respected practitioners in the field” [49]. A national No-Fault COVID-19 Indemnity Scheme for moderate to severe adverse events related to COVID-19 vaccines has been implemented by the Australian Government [50].

#### 3.5.2. Indonesia

Medical liability in Indonesia is based on tort law, as influenced by the medical liability laws in force during the years of the Dutch colonial administration. Civil and criminal medical liability are established by the Indonesian law. This responsibility can be a deviation from the standards of care or from the ethical code to which healthcare professionals must adhere to. Civil liability applies if a casual relation is found between the medical conduct (malpractice or mild negligence) and financial damage. Criminal liability is established for negligence if the actions of a medical practitioner cause death to a person; if the negligence is proved, the criminal offense is punishable with imprisonment. Lastly, ethical malpractice is established by the Indonesian Code of Ethics, in accordance with the Declaration of Helsinki and the Geneva Declaration, comprehending twenty-eight disciplinary violations [51]. Vaccination against COVID-19 is currently mandatory in Indonesia [52]. Interestingly, various Indonesian laws approved before the COVID-19 pandemic established that, in the case of an infectious disease outbreak, the right to refuse medical intervention does not apply, and vaccinations can be made mandatory [53].

#### 3.5.3. Papua New Guinea

In Papua New Guinea, the medical negligence is influenced by the English and the Australian Common Law. It is in civil courts that damages can be granted to a plaintiff if the medical doctor is found negligent. Doctors need to have reasonable skill and judgement, meaning that medical professionals must not be careless, negligent, or fail to meet standards of care. Furthermore, negligence requires the medical professional’s actions to be causal to the outcome. (i.e., if a doctor is found in breach of the duty of care, but the patient would have had the same outcome either way, the causal link does not subsist). The State may be liable if the negligence was committed by an employee of the State (i.e., hospital worker) during the course of their employment; however, Papua New Guinea courts are reticent to grant exemplary damages if the State could be held liable [54]. Vaccination against COVID-19 is not mandatory in Papua New Guinea, although the Prime Minister declared that, as far as workplaces are concerned, employers can establish internal policies regarding vaccination to protect their employees if they see fit [55].

### 3.6. South America

#### 3.6.1. Brazil

Brazil’s medical liability system is principally based on civil liability. Specifically, liability arises because of the breach of a contract that provides for professional obligations between the patient and the doctor/healthcare facility. This is the obligation to compensate for damages caused in the practice of medicine as a result of an error, configured, apart from specific cases, as an obligation of means and not of result. However, the doctor must follow the guidelines and good practices of the service with diligence, otherwise liability will arise.

The medical profession is regulated by Resolution No. 2217/2018 of the Federal Council of Medicine, which approved the Code of Ethics for Physicians. Article 1 of the Resolution states that the doctor is prohibited from “causing harm to the patient, by acts or omissions considered as negligence, imprudence or inexperience”. Article 1 adds that “medical liability is always personal and cannot be presumed”. In other words, it confirms the physician’s subjective liability arising from the obligation of means [56]. However, the doctrine argues that some activities may constitute an obligation of result, such as plastic surgery [57].

#### 3.6.2. Colombia

In Colombia, medical liability is divided into different areas of applicability: ethical, penal, civil, administrative, and disciplinary [58]. In the ethical field, the doctor is liable in accordance with Law No. 23 of 1981 [59,60]. Criminal liability is personal, with the doctor being judged by the Fiscalía General de la Nación (a judicial public power body with full administrative and budgetary autonomy, whose function is to provide citizens with complete and effective administration of justice) [61]. In Civil and Administrative Law, the damage suffered by the patient is economically compensated, although in Administrative Law the case is brought directly against the public health facility that compensates the damage [58].

#### 3.6.3. Argentina

The Doctors in Argentina are subject to the civil liability regime established by the Civil and Commercial Code (CCC). The medical liability arises when the duty of care is violated, as set out in Article 1768 of the CCC (i.e., the duty to perform his or her service with diligence, according to the technical knowledge and methods proper to the specific area of competence). In most medical malpractice cases also the hospital is sued and if the doctor’s negligence is proven, the courts presume that the hospitals were also negligent in their duty to provide adequate medical care; therefore, they can also be liable for damages inflicted on patients [62].

## 4. Discussion

Our study evaluated the medical liability system in 18 countries worldwide and the enactment of specific laws to protect the activities of the HCWs during the pandemic, especially inherent in the administration of COVID-19 vaccines. We found that in seven countries (Nigeria, India, Pakistan, USA, Canada, Australia, and Papua New Guinea) the medical liability system is based on Common Law, while in eleven (Ethiopia, Egypt, China, Russian Federation, Turkey, Germany, Mexico, Indonesia, Brazil, Colombia, and Argentina) it is mainly based on the Civil Law system. The Civil Law model, currently the most widespread in the world, is based on written law and the decisive role of the law. In particular, Civil Law systems are based on the “codification” of the law characterized by being general and abstract: they do not analyze the concrete fact but regulate general hypotheses from which individual cases will then have to be extrapolated. This means that law has a pre-eminent role in guiding the decisions of the judiciary, which must adhere to it on a case-by-case basis [73].

On the other hand, Common Law is based mainly on judges’ decisions. Indeed, judgments are binding regarding future cases based on the so-called stare decisis principle, according to which what binds the judge are the judicial precedents on the matter; i.e., the judgments. Thus, written law is less relevant, assuming consequently a secondary role. Regarding HCWs, this results in stricter decisions if the same actions were found guilty in a previous instance [74].

Considering the application of specific laws to protect HCWs who vaccinate during the pandemic, among the countries analyzed in our study only the USA and Canada provided HCWs with a liability exemption for claims of loss caused by the administration or by the use of medical countermeasures to the COVID-19 pandemic (including vaccines). The guiding principle for this specific type of policy is to ensure that health professionals can devote themselves exclusively to their work and the care of patients, without the fear of future claims [75]. This also facilitates the doctor–patient relationship and limits the phenomenon of defensive medicine avoiding dispersion of resources, especially in times of crisis [76].

While liability policies are established as guaranties for HCWs, compensation programs aim to offer guaranties for adverse events caused by vaccination. Since the inception of VICPs, considerable amounts of compensation have been provided for vaccine injuries in various countries through their respective programs. These programs are crucial for the increase in public trust in immunization, especially for COVID-19 [77]. Indeed, the pandemic stressed the impact of misinformation and fake news related to vaccines safety and efficacy on vaccination campaigns. Furthermore, the COVAX No-Fault Compensation Program for Advance Market Commitment for Eligible Economies is fundamental. It is endorsed by the WHO and it focuses on low- and middle-income countries [11]. Moreover, many compensation mechanisms already in place in countries have been activated to ensure compensation from COVID-19, even if in some countries it is unclear whether these funds can be used only for a specific type of vaccination (mandatory or recommended) and by age (pediatric or adult) [77]. Thus, governments have to clarify these issues as a priority in order to further ensure public confidence and security towards vaccination.

Furthermore, by examining the introduction of compulsory vaccination, both countries that introduced new medical liability policies for vaccinating HCWs (USA and Canada) enforced mandatory vaccination on selected groups of workers (see Table 1 for details). However, six other countries included in the review implemented mandatory vaccination policies (Nigeria, Russian Federation, Turkey, Germany, Australia, and Indonesia) despite none of them introducing policies for vaccinating HCWs. Therefore, mandatory vaccination policies did not appear to influence the introduction of new medical liability policies pertaining to vaccination during the COVID-19 pandemic.

This review highlights the high variability among the included countries concerning vaccination coverage; it is important to note that these differences might have influenced policymakers. Indeed, countries with a low vaccination coverage (i.e., Papua New Guinea, Nigeria, and Ethiopia—see Table 1 for vaccination coverage details) might have been less inclined to introduce a liability shield for HCWs, and this is supported by our results. On the other hand, among those countries with a high vaccination coverage (i.e., China, Canada, Argentina, and Australia—see Table 1 for vaccination coverage details), only in the USA and in Canada new medical liability policies were introduced, which leads to the assumption that the extent of the problem is different in Western and Eastern countries; for instance, China, the country with the highest vaccination rate, did not introduce any medical liability policies.

This review has some strengths and limitations. We caution against a possible source of bias given the languages considered in the literature search, since it is possible that minor policies would not have been as easy to find in English or that details were not reported. However, we extensively searched the literature of each of the included country to overcome this possible bias, using several different sources of information and not excluding evidence not primarily in English. Furthermore, we focused only on the three most populous countries on each continent; thus, we may have excluded countries that actually passed specific laws to protect vaccinating HCWs. On the other hand, our study represents the first attempt on a global scale to assess the adoption of laws to protect HCWs during the pandemic and to analyze the medical liability legislative system applied in different countries.

## 5. Conclusions

From an international perspective, our results highlight great variability among the included countries in terms of vaccination coverage and compulsory vaccination policies. Despite the similarities in the medical liability systems—mainly based on Civil and Common Law—many facets were highlighted by our review in the countries included. However, the trait d’union is represented by the specific policies issued to reshape the medical-legal litigation, guaranteeing protection for the population and attempting to prevent the phenomenon of defensive medicine. In addition to offering a valuable tool for policymakers, our work highlights the need for pre-pandemic contingency plans, emphasizing the importance of preparedness.

Given the global burden of COVID-19 and the similar challenges that it posed to different countries, this review highlights the need for a coordinated and integrated intervention across countries, to align medical liability policies during the COVID-19 pandemic.

## Figures and Tables

**Table 1 vaccines-10-01275-t001:** COVID-19 vaccination: law and policies, mandatory vaccination, vaccine injury compensation program, and vaccination coverage.

Continent	Country *	Vaccination Coverage **	COVID-19 Compulsory Vaccination	COVID-19 Vaccine Injury Compensation Programs	New Law/Policies for COVID-19 Vaccinating HCWs
**Africa**	Nigeria	13.38 (4.11)	Federal Government employees [63]	WHO COVAX No-fault Compensation Program [11]	None
	Ethiopia	22.77 (15.71)	No	WHO COVAX No-fault Compensation Program [11]	None
	Egypt	108 (69.73)	No	WHO COVAX No-fault Compensation Program [11]	None
**Asia**	China	213.28 (83.98)	No	Article 56 of the Vaccine Administration Law [64]	None
	India	128.75 (57.42)	No	WHO COVAX No-fault Compensation Program [11]	None
	Pakistan	96.67 (45.31)	No	WHO COVAX No-fault Compensation Program [11]	None
**Europe**	Russian Federation	110.2 (49.06)	All workers with public-facing roles in Moscow; People over 60 years old or with chronic illness in St. Petersburg [65]	None	None
	Turkey	171.7 (63.39)	Civil servants, including teachers and healthcare workers [65]	None	None
	Germany	203.3 (75.15)	People over 18 years old [65]	Germany no-fault compensation program (Federal Communicable Diseases Act + Federal Social Assistance Law) [66]	None
**North America**	USA	162.41 (63.5)	All federal workers, contractors, private sector workers in companies with ≥100 employees, public-sector workers [67]	National Vaccine Injury Compensation Program and Countermeasures Injury Compensation Program [68,69]	Public Readiness and Emergency Preparedness Act [38]
	Canada	212.58 (81.33)	All federal workers [65]	Canadian Vaccine Injury Support Program [70]	COVID-19 Emergency Response Act (British Columbia: Bill 70; Ontario: Bill 218) [43,44,71]
	Mexico	139.04 (61.08)	No	None	None
**Oceania**	Australia	211.94 (80.3)	For some occupational categories (aged-care workers) [72]	WHO COVID-19 No-fault Compensation Program [11]	None
	Indonesia	126.39 (52.83)	People over 18 years old [52]	WHO COVAX No-fault Compensation Program [11]	None
	Papua Nuova Guinea	4.61 (2.8)	No	WHO COVAX No-fault Compensation Program [11]	None
**South America**	Brazil	173.85 (69.99)	No	WHO COVAX No-fault Compensation Program [11]	None
	Colombia	150.8 (65.58)	No	WHO COVAX No-fault Compensation Program [11]	None
	Argentina	205.79 (80.18)	No	WHO COVAX No-fault Compensation Program [11]	None

* The continents are sorted alphabetically (considering North and South America separately) for each continent, the three countries with the highest population are sorted in descending order (data from the United Nations database https://data.un.org/ (accessed on 11 March 2022)). ** Expressed as total vaccine doses administered per 100 population/(persons fully vaccinated with last dose of primary series per 100 population), at 11March 2022.

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
