# Peer review of "COVID-19 Vaccination and Medical Liability: An International Perspective in 18 Countries"

_vaccines, 2022, doi:10.3390/vaccines10081275_

Round 1

Reviewer 1 Report

1. Although authors covers countries from each continent, there are 18 countries where have been reviewed. Hence, this manuscript studied 18 counties, not global setting. Authors may change the title or add some explanation.

2. There seem three methods to identify relevant articles, howeger, description on the way to identify country illgeal framework is a sort of missing. Describe it with consise

3. vaccine coverage is interesting, while authors just stated high/low. Depend on the countries, a taragetted proportion of vaccine aministration would be changes. Authos can discuss this section more.

Author Response

Reviewer #1

  1. Although authors covers countries from each continent, there are 18 countries where have been reviewed. Hence, this manuscript studied 18 counties, not global setting. Authors may change the title or add some explanation.

Thank you for pointing out the lack of clarity in the title of the manuscript. We agree that the term 'global' can be confusing and does not correspond to the countries actually analysed. Therefore, we change the title to “COVID-19 vaccination and medical liability: an international perspective in 18 countries”. 

  1. There seem three methods to identify relevant articles, howeger, description on the way to identify country illgeal framework is a sort of missing. Describe it with consise

In outlining the framework of medical liability, we have adopted three methods to try to avoid the limitations of each. Indeed, it is possible that the literature is not up-to-date on the most recent changes in laws and regulations or that some policies are only published in the original language of the country. Therefore, we have resorted to grey literature and national official archives to cope with these main limitations. Moreover, medical liability is usually based on a composite set of rules, consisting of civil, criminal and administrative jurisdiction. It is possible that the relevant articles identified with the search were not able to provide a sufficiently complete picture. We have added the following paragraph to the text, hoping to better clarify why we have adopted this methodology: “We combined the results from the three different sources to outline the framework of medical liability in each country, emphasising, where appropriate, the peculiar changes that occurred during the pandemic and related to vaccinating doctors.”

  1. vaccine coverage is interesting, while authors just stated high/low. Depend on the countries, a taragetted proportion of vaccine aministration would be changes. Authos can discuss this section more.

Thank you for this comment. The vaccination coverage rates for individual countries are reported in Table 1. As far as the discussion is concerned, we have only classified coverage rates as high/low to use this data to discuss medical liability policies for vaccinating healthcare personnel – which is the focus of the manuscript – and its relation to vaccination coverages, but as per your valuable suggestion we have amended the paragraph as follows: “countries with a low vaccination coverage (i.e.: Papua New Guinea, Nigeria, Ethiopia. See Table 1 for vaccination coverage details) [...] countries with a high vaccination coverage (i.e.: China, Canada, Argentina, Australia. See Table 1 for vaccination coverage details)."

Reviewer 2 Report

Article: COVID-19 vaccination and medical liability: a global perspective

Article presentation – General information

The study is an original study that presented the among the three most populous countries of each continent, which medical liability system is in place, and if during the COVID-19 pandemic there have been introduced vaccine injury compensation programs or laws or new policies to protect vaccination health care workers. In addition, the authors aimed to investigate whether vaccination is mandatory for specific occupational groups or the general population.

Article classification

Minor comment: The authors revised the medical liability system in three countries of each continent. However, in my opinion, the authors should classify the study as original ones.

Title: 

The use of “global perspective” can be considered an error. The authors only used the information for each continent's three most populous countries. The study is limited to generalizing its findings to other countries, mainly those with a low number of citizens and completely different characteristics, such as health support management, health conditions, financial support, and others. 

Abstract

To avoid using the sentence “… to defeat the virus”. Vaccination is the best approach to control the COVID-19 pandemic. However, the vaccination was not able to “… defeat the SARS-CoV-2”.

Conclusion: To revise the following excerpt: “From a global perspective, our results highlight great variability across the included countries regarding COVID-19 related policies and legal systems for medical liability.” – based on the findings presented by the authors, I believe there is a low variability across the included countries regarding COVID-19 related policies and legal systems for medical liability. For example:

i-) In seven countries, the medical liability system was based on Common Law, while in eleven, it is mainly based on the Civil Law;

ii-) only the USA and Canada provided immunity from liability;

iii-) fourteen countries have adopted compensation funds.

Keywords: The authors can cite the keywords using alphabetical order.

Introduction

The authors use Italy as a policy example. However, it is important to contextualize other international scenarios. In addition, reading the text, I could not identify the results and the discussion for the second study objective: "Finally, it is also aimed at investigating whether vaccination is mandatory for specific occupational groups or the general population.”

Methods

The authors included the three countries with the highest population for each continent. It is also important to have other countries, such as those with the lowest population for each continent.

The data search (Medical liability research topic) was based on PubMed and Google for articles. It also performed a search in the country’s institutional repositories. However, I noticed that some papers were not cited in the article and discussed the study topic. In addition, several health measures were performed in the countries included in the study, and these measures were published only in the mother language of these countries. In brief, the study has this vital limitation in identifying each measure adopted for each country.

The authors should describe using more detail the procedures to identify the presence of mandatory vaccination and compensation programs in each country.

Results

The authors should improve the information for the changes included based on the impact of the COVID-19 pandemic. Also, in cases of no changes, the authors should better describe the indirect evidence of COVID-19 in the medical liability.

Discussion

It is essential to include a comparison between the different countries in the discussion. In addition, in my opinion, the authors should include other examples of countries in the results and discussion sections, such as those with the lowest number of citizens in each continent. 

Conclusions

The authors should rewrite the conclusion based on the study's limitations. In my opinion, the authors should avoid using the term “global perspective.”

Author Response

Reviewer #2 

Article classification

Minor comment: The authors revised the medical liability system in three countries of each continent. However, in my opinion, the authors should classify the study as original ones.

Thank you for your comment; however, considering the methodology we used to collect data, we would like to maintain the manuscript as a narrative review.

Title: 

The use of “global perspective” can be considered an error. The authors only used the information for each continent's three most populous countries. The study is limited to generalizing its findings to other countries, mainly those with a low number of citizens and completely different characteristics, such as health support management, health conditions, financial support, and others. 

Thank you for pointing out the lack of clarity in the title of the manuscript. We agree that the term 'global' can be confusing and does not correspond to the countries actually analysed. Therefore, we change the title to “COVID-19 vaccination and medical liability: an international perspective in 18 countries” 

Abstract

To avoid using the sentence “… to defeat the virus”. Vaccination is the best approach to control the COVID-19 pandemic. However, the vaccination was not able to “… defeat the SARS-CoV-2”.

Conclusion: To revise the following excerpt: “From a global perspective, our results highlight great variability across the included countries regarding COVID-19 related policies and legal systems for medical liability.” – based on the findings presented by the authors, I believe there is a low variability across the included countries regarding COVID-19 related policies and legal systems for medical liability. For example:

i-) In seven countries, the medical liability system was based on Common Law, while in eleven, it is mainly based on the Civil Law;

ii-) only the USA and Canada provided immunity from liability;

iii-) fourteen countries have adopted compensation funds.

Thank you for your comment, the sentence "the most effective measure to defeat the virus” was changed to: "the most effective prevention measure". Furthermore, the last sentence has been replaced with: " From an international perspective, our results highlight that in eleven (61.1%) countries medical liability is mainly based on Civil Law, whilst in seven (38.9%) it is based on Common Law" to give a more precise account of our results.  

Keywords: The authors can cite the keywords using alphabetical order.

The keywords have been organized in alphabetical order: "Compensation programs; COVID-19 vaccination; legal system; legislation; Medical liability"

 Introduction

The authors use Italy as a policy example. However, it is important to contextualize other international scenarios. In addition, reading the text, I could not identify the results and the discussion for the second study objective: "Finally, it is also aimed at investigating whether vaccination is mandatory for specific occupational groups or the general population.”

 As per your valuable suggestion, in the introduction we amended the paragraph provided as example as follows: “In Europe, for example, France and Italy have resorted to these instruments, establishing respectively a specific compensation system for COVID-19 vaccine-related injuries for the former and a criminal shield for the latter.”

Thank for your valuable comment. At the end of the introduction, we rephrased the article’s objective, changing this particular sentence in the introduction as follows: “[...] providing also information whether, in the selected countries, vaccination is mandatory for specific occupational groups or the general population.”Furthermore, in the discussion we added the following paragraph concerning our results on mandatory vaccination: “Furthermore, examining the introduction of compulsory vaccination, both countries that introduced new medical liability policies for vaccinating HCWs (USA and Canada) enforced mandatory vaccination on selected groups of workers (see Table 1 for details). However, six other countries included in the review implemented mandatory vaccination policies (Nigeria, Russian Federation, Turkey, Germany, Australia, and Indonesia) despite none of them introducing policies for vaccinating HCWs. Therefore, mandatory vaccination policies did not appear to influence the introduction of new medical liability policies pertaining to vaccination during the COVID-19 pandemic.”

Methods

The authors included the three countries with the highest population for each continent. It is also important to have other countries, such as those with the lowest population for each continent.

The data search (Medical liability research topic) was based on PubMed and Google for articles. It also performed a search in the country’s institutional repositories. However, I noticed that some papers were not cited in the article and discussed the study topic. In addition, several health measures were performed in the countries included in the study, and these measures were published only in the mother language of these countries. In brief, the study has this vital limitation in identifying each measure adopted for each country.

The authors should describe using more detail the procedures to identify the presence of mandatory vaccination and compensation programs in each country.

In outlining the framework of medical liability, we have adopted three methods to try to avoid the limitations of each. Indeed, it is possible that the literature is not up-to-date on the most recent changes in laws and regulations or that some policies are only published in the original language of the country. Therefore, we have resorted to grey literature and national official archives to cope with these main limitations. Moreover, medical liability is usually based on a composite set of rules, consisting of civil, criminal and administrative jurisdiction. It is possible that the relevant articles identified with the search were not able to provide a sufficiently complete picture. We have added the following paragraph to the text, hoping to better clarify why we have adopted this methodology: “We combined the results from the three different sources to outline the framework of medical liability in each country, emphasising, where appropriate, the peculiar changes that occurred during the pandemic and related to vaccinating doctors.” In addition, it is possible that a few relevant articles were not included, as this review was framed as narrative and not systematic, our aim was not to report all scientific publications on the topic, but to frame the medical liability scenario of the included countries, as well as mandatory vaccination laws and compensation programs: this was, as stated, performed through a search in institutional repositories. Whereas it is true we might have missed policies emanated in a language different than English, this was highlighted as a study limit, as follows: "One of the study’s limitations is the language, as each country included in the review has its legislation written in its own language, and, although the key messages are reported in this review, some details may have been lost in translation".

Results

The authors should improve the information for the changes included based on the impact of the COVID-19 pandemic. Also, in cases of no changes, the authors should better describe the indirect evidence of COVID-19 in the medical liability.

Thank you for this comment, we appreciate the suggestion. However, it is very difficult to quantify the impact that COVID-19 has had on the included countries in general; in the review, we do not take into account COVID-19 cases or deaths, we do not analyze direct and indirect healthcare and non-healthcare costs that COVID-19 cases have caused to each country, or the social impact of the pandemic. Our only “metric" is vaccination coverage, because we did not aim to evaluate the impact of COVID-19 or its indirect effects on medical liability. Our only medical liability outcome specifically pertains to new policies for healthcare workers administering the vaccination, which we researched extensively. We feel that adding indirect effects of the pandemic on medical liability (i.e.: Brazil has introduced a telemedicine liability policy, but many more have been introduced across these 18 countries) would shift the focus from our main scope. However, your precious comment gives us an amazing input to keep researching into this field and we hope future research can answer the query of indirect effects the pandemic has had on medical liability in its complex entirety.

Discussion

It is essential to include a comparison between the different countries in the discussion. In addition, in my opinion, the authors should include other examples of countries in the results and discussion sections, such as those with the lowest number of citizens in each continent. 

While thanking you for your valuable suggestion, we think that further research on this topic is needed and certainly taking more countries into consideration would undoubtedly enrich the comparison. Of course, it is possible that the less populous countries differ greatly and show greater complexity in a 'true' global perspective. However, in this paper we have focused on the three most populous countries on each continent as a proxy for the continent itself. In fact, in this way we have considered the largest population on an international scale, allowing us to analyse in greater depth the legal system of each country considered, vaccination coverage, specific policies implemented during the pandemic and vaccination obligations. For reasons of feasibility and to better serve the aim of the work (which is to provide a framework and then a comparison), we have limited the selection of countries. In some cases, we believe that their analysis does not add particular value, e.g. in Europe, the least populated countries are Liechtenstein, San Marino, Vatican City; in Africa they are Sao Tome and Principe and Seychelles; in Asia and America they are mostly islands. Therefore, the methodology should have been modelled in such a way as to avoid selecting very small countries, thus adopting different evaluation and selection measures than those stated for the most populous countries. Although this is a narrative review, we preferred to be rigorous and keep the selection method as simple and clear as possible.

Conclusions

The authors should rewrite the conclusion based on the study's limitations. In my opinion, the authors should avoid using the term “global perspective.”

We changed from “global perspective” to “international perspective”. Furthermore, in agreement with your comment, we changed the conclusions paragraph as follows: " From an international  perspective, our results highlight great variability across the included countries, in terms of COVID-19 related policies and of the legal system for medical liability in terms of vaccination coverage and compulsory vaccination policies. Despite the similarities in the medical liability systems - mainly based on Civil and Common Law - many facets were highlighted by our review in the countries included."

Round 2

Reviewer 2 Report

I read with great attention the revision performed by the authors. I believe that the authors improved some topics of the manuscript or discussed the limitations to correct the manuscript as requested during the review. I thank the authors' efforts; besides that, my primary concerns about the study were not solved. For example, the authors did not include the information for other countries (such as those with the lowest number of citizens). Also, the study has a critical bias: “One of the study’s limitations is the language, as each country included in the review has its legislation written in its own language, and, although the key messages are reported in this review, some details may have been lost in translation" which can compromise the findings of the study.

Author Response

Reviewer comments

I read with great attention the revision performed by the authors. I believe that the authors improved some topics of the manuscript or discussed the limitations to correct the manuscript as requested during the review. I thank the authors' efforts; besides that, my primary concerns about the study were not solved. For example, the authors did not include the information for other countries (such as those with the lowest number of citizens). Also, the study has a critical bias: “One of the study’s limitations is the language, as each country included in the review has its legislation written in its own language, and, although the key messages are reported in this review, some details may have been lost in translation" which can compromise the findings of the study.

Response

Dear Reviewer,

Thank you for pointing out that the highlighted limitation might be misinterpreted, and we propose change the sentence (lines 463-468) “One of the study’s limitations is the language, as each country included in the review has its legislation written in its own language, and, although the key messages are reported in this review, some details may have been lost in translation” with "We revise as a possible source of bias the languages considered in the search, since it is possible that minor policies would have not been as easy to find in English or that details were not reported. However, we extensively searched for each of the included country to overcome this possible bias, using several different sources of information and not excluding found evidence even if not primarily in English." In fact, during the research the finding of results in the original language was not a reason for exclusion. We recurred to translation when necessary and compared different sources of information to avoid any evitable gap. However, the results were in most cases in English. Nevertheless, we recognize that having set up the search in English may have led to the loss of some minor information and we consider this a limitation of the study. We also recognize that a further limitation of the study is the choice of countries considered. We do not believe we can say that three countries, albeit the most populous, represent the entire continent. While this may be close to reality for North America and Oceania, it certainly is not for Europe and Africa. In fact, Russia, Turkey and Germany are certainly not representative of the complexity represented by the 57 countries in the European region (however, we cited a specific work with a similar topic in ten European Countries). We have chosen to consider the three most populous countries per continent for two main reasons. Firstly, thanks to this choice we are able to take into account as large a share of the world's population as possible, taking care not to cut out entire areas of the planet. Out of completeness, we accounted for the 63% world share of population, without leaving out any continent (which would have happened if we just went by population). In addition, the whole of Oceania counts for 0.53 percent of the world's population share, and the three countries considered cover 0.50 percent. This means that all the “excluded” countries count for 0.03% altogether.

And here is the second reason behind our choice. In designing the study, we thought about possible strategies to grant the best value to the work. Of course, this is just our opinion, and we can’t see how the addition of more countries, no different from those already included, could add value to the work. Moreover, studies and models aimed to establishes the relationship between population and regulation, show how population was not thought to be very important in determining government activities and even more populous states regulate more activities, and these regulations contain greater detail than those of less populous states (e.g. Casey B. Mulligan & Andrei Shleifer, 2004. "Population and Regulation," NBER Working Papers 10234, National Bureau of Economic Research, Inc.) To be more precise, even if we are keen on emphasizing that we do not offer a comparison between different countries within the same continent, we offer a broad and inclusive landscape; in fact, we evaluate low- and high-income countries with varying vaccination coverage and different populations. We consider, in fact, China (which is the most populous one with about 20% of world share) as well as Papua New Guinea (0,11% world share). Therefore, we believe that adding countries, as much as it may enrich the offered landscape with details, does not compensate for a methodological shortcoming since the countries represented offer variability in terms of all the characteristics considered (population vaccination coverage legal system medical-legal responsibility policies implemented mandatory vaccination or not). In our opinion, adding more countries would only serve the scope of compare each continent internally (or in other words, it would give the possibility of comparing most populous countries versus less populous ones in each continent). However, this is not the scope of this research. Adding the three countries with the lowest populations per continent or adding more countries for greater representativeness of each continent, is not the purpose of this research, where the selection of countries offers only an international perspective as COVID-19 pandemic has been, and still is, an international phenomenon. And from what the results show, there is no correlation in the countries analyzed between the implementation of new COVID-19 related policies and the pre-existing medical legal liability system nor the vaccination coverage offered an interpretation bias.

Certainly, it is possible that other choice systems could have provided a better overview than the one presented. However, we believe the purpose of the review is to provide an overview of several states, not to make a worldwide per-continent comparison between larger and smaller countries. Complying with the request would mean starting from the very definition of our scope and then adjust methods and results accordingly. We are extremely thankful for your inputs, and we will deploy other methods of representativeness for future works.
